# Characterization of *Clostridioides difficile* Persister Cells and Their Role in Antibiotic Tolerance

**DOI:** 10.3390/microorganisms12071394

**Published:** 2024-07-09

**Authors:** Osvaldo Inostroza, Juan A. Fuentes, Paulina Yáñez, Giovanni Espinoza, Omar Fica, Camila Queraltó, José Rodríguez, Isidora Flores, Ruth González, Jorge A. Soto, Iván L. Calderón, Fernando Gil

**Affiliations:** 1Microbiota-Host Interactions and Clostridia Research Group, Departamento de Ciencias Biológicas, Facultad de Ciencias de la Vida, Universidad Andres Bello, Santiago 8370186, Chile; osv.inostroza.t@gmail.com (O.I.); pauli.fyc@gmail.com (P.Y.); esgio01@gmail.com (G.E.); o.fica.sch@gmail.com (O.F.); camilabelen.-@hotmail.com (C.Q.); j.rodrguezbernal@uandresbello.edu (J.R.); i.floresrivera@uandresbello.edu (I.F.); ruth.gonzalezm@gmail.com (R.G.); 2Laboratorio de Genética y Patogénesis Bacteriana, Departamento de Ciencias Biológicas, Facultad de Ciencias de la Vida, Universidad Andres Bello, Santiago 8370186, Chile; jfuentes@unab.cl; 3Millennium Institute on Immunology and Immunotherapy, Departamento de Ciencias Biológicas, Facultad de Ciencias de la Vida, Universidad Andres Bello, Santiago 8370186, Chile; jorge.soto.r@unab.cl; 4Laboratorio de RNAs Bacterianos, Departamento de Ciencias Biológicas, Facultad de Ciencias de la Vida, Universidad Andres Bello, Santiago 8370186, Chile; 5ANID-Millennium Science Initiative Program-Millennium Nucleus in the Biology of the Intestinal Microbiota, Santiago 8370186, Chile

**Keywords:** *Clostridioides difficile*, persister cells, antibiotic tolerance

## Abstract

*Clostridioides difficile* is a Gram-positive pathogen known for its toxin production and spore formation. It is primarily responsible for most cases of antibiotic-associated diarrhea. Bacterial persisters are a small subset of the population that exhibits transient tolerance to bactericidal substances, and they are of significant medical concern due to their association with the emergence of antibiotic resistance and difficult-to-treat chronic or recurrent infections. Vancomycin, the predominant antibiotic utilized in the management of *C. difficile* infection, is extensively applied in the realm of clinical practice. Previous studies have demonstrated a persister-like phenotype with treatments involving this antibiotic. However, the mechanism in *C. difficile* remains largely unknown, primarily due to the challenge of isolating this small population at any given time. To better characterize *C. difficile* persister cells, we present a study that enables the enrichment and characterization of persister cells from bacterial cultures in both the exponential and stationary phases. Moreover, we could differentiate between triggered (induced using antibiotics such as vancomycin) and spontaneous (stochastic) persister cells. Additionally, we observed the involvement of toxin-antitoxin systems and Clp proteases in persister cell formation.

## 1. Introduction

*Clostridioides difficile*, an anaerobic Gram-positive pathogen [1], is notorious for causing a spectrum of gastrointestinal ailments, including diarrhea, toxic megacolon, and pseudomembranous colitis [2]. In healthcare environments, *C. difficile* infection (CDI) is strongly correlated with antibiotics, which disrupt the normal colonic bacterial population, leading to dysbiosis. This disruption affects the metabolism of bile salts, particularly taurocholate, which is crucial for *C. difficile* spore germination [3,4,5,6,7,8]. The mortality rate associated with CDI typically hovers around 5%, but during severe outbreaks, it can spike to 20% [6]. Furthermore, CDI recurrence poses a substantial challenge in management, with up to 35% of patients experiencing a recurrence after their initial episode [9] and the likelihood of recurrence escalating with subsequent episodes. Recurrence may involve either the relapse of the same strain or reinfection with a different strain, occurring in approximately 38–56% of cases [3,4,5,6,7,8]. A key contributor to recurrent disease is *C. difficile*’s capacity to re-sporulate and persist in the gastrointestinal tract, even in the presence of antibiotics [9]. Current guidelines and FDA approval lists offer multiple treatment options (vancomycin, fidaxomicin, and metronidazole) and possible adjunctive therapies (antitoxin antibodies, FMT (fecal microbiota transplantation), among others) [10], but vancomycin and fidaxomicin are the current standard treatments for CDI [11]. Vancomycin, belonging to the class of tricyclic glycopeptides, inhibits cell wall synthesis by binding to the D-alanyl-D-alanine ends of cell wall precursors [12].

Recent investigations suggest that the outcomes of CDI may not solely correlate with virulence characteristics or sporulation levels, indicating the involvement of other biological processes [2]. The bacterium’s persistence in biofilms could underlie colonization and relapse, potentially facilitating adherence to colonic epithelial cells [13,14,15,16], in which sporulation and persister cell formation could be different mechanisms for pathogen persistence occurring at different stages of the infectious cycle, serving different purposes. Sporulation is a complex and gradual process that produces a resilient morphotype, the spore, adapted for transmission between hosts or as a long-term reservoir between recurrent infections [3,4,5,6,7,8]. In contrast, persister cells are generated more rapidly, which is particularly suited for antibiotic tolerance during biofilm formation [17,18,19,20,21,22,23,24].

It has been hypothesized that two types of persister cells exist. Triggered persister cells (or type I) are induced by environmental stimuli, such as antibiotic use, while spontaneous (or type II) persister cells arise stochastically during bacterial growth; these persister cells accumulate during growth and are typically found in the stationary phase [21,22,23,24]. Since Balaban defined persistence to antibiotics in 2019 [25], triggered persistence has been observed in bacteria as a response to stress signals like starvation, where cells enter a persistent state even after the stress is removed. Various stressors can trigger this persistence, including nutrient limitation and antibiotic exposure. Spontaneous persistence, on the other hand, happens without an obvious trigger during stable growth conditions and remains constant [26]. Despite recent advances in the field, the mechanisms responsible for persister cell formation in *C. difficile* are still largely unknown. This lack of understanding can be attributed, to a significant extent, to the challenges associated with isolating the exceedingly small fraction of the population that finds itself in this state at any given point in time, as well as the anaerobic characteristics of this pathogen. In this study, we present, to our knowledge, the first investigation that enables the enrichment of *C. difficile* persister cells and allows for their thorough characterization. We have successfully differentiated between two distinct types of persister cells, namely triggered persister cells induced by vancomycin during both the exponential and stationary phases and spontaneous persister cells induced by stochastic processes. Additionally, we identified RNA metabolic-decreased persister cells in *C. difficile*. In addition to these findings, we also observed the overexpression of toxin–antitoxin systems and Clp protease genes, which are known to play pivotal roles in forming persister cells in several bacteria [27,28,29,30,31].

## 2. Methods

**Bacterial culture:** The growth of *Clostridioides difficile* R20291 [32,33] and Δ*spo0A* strains were performed in BHIS (BD difco, San Jose, CA, USA), containing 0.2% taurocholate (Merck KGaA, Darmstadt, Germany), 0.1% fructose (Merck KGaA, Darmstadt, Germany), and 0.1% glucose (Merck KGaA, Darmstadt, Germany). This growth occurred at 37 °C under anaerobic conditions within the BACTRON EZ anaerobic chamber provided by Shellab, Cornelius, USA, as shown in Appendix A. To obtain cultures in the exponential and stationary phases, an overnight (16 h) culture was utilized. For exponential growth, 1% *v*/*v* of this culture was used until OD_600_ 0.5 or OD_600_ 0.9 for the stationary culture. To isolate the persister cells, vancomycin Merck KGaA, Darmstadt, Germany, (100 μg/mL), ceftriaxone Merck KGaA, Darmstadt, Germany (50 μg/mL), pefloxacin Merck KGaA, Darmstadt, Germany (50 μg /mL), or rifampicin Merck KGaA, Darmstadt, Germany (10 μg/mL) (10× MIC for each antibiotic) were administered separately for 1 h.

**Enrichment of persister cells.** This procedure involved subjecting exponential or stationary cultures to a lysis solution (1 mM NaOH, Merck, and 0.005% SDS, Merck) for 10, 20, 30, and 60 min [21] followed by 4 mg/mL of lysozyme (Sigma-Aldrich, St Louis, MO, USA) for 15 min (0.5× MIC for *C. difficile*) [26]. Subsequently, aliquots were plated at 2, 10, 15, 30, and 60 min. The identical experimental procedure was also applied to the cultures that underwent antibiotic treatment, as previously described, but with a duration of 2 h. Plated aliquots were then collected at 2, 10, 30, and 60 min after the lysis treatment (see Appendix A).

**Staining of persister cells.** As previously described, under an anaerobic chamber, the culture enriched in persister cells was washed twice using 1× TE buffer [33,34]. To this culture, Propidium Iodide Sigma-Aldrich (40 ng/mL) and Thioflavin-T (Thermoscientific, Waltham, MA, USA) (5 µg/mL, 0.008× of MIC for *C. difficile* determined for this work) were added and incubated for 15 min in complete darkness. Thioflavin-T only stained the intact bacteria with metabolically active RNA but no lysed bacteria. Two additional washes using 1× TE buffer were performed following the incubation period. Subsequently, the washed cell suspension that had been stained was observed using an Olympus™, Tokyo, Japan epifluorescent microscope equipped with FITC (fluorescein isothiocyanate with a center wavelength of 475 nm and FWHM of 35 nm) and TRITC (Tetramethylrhodamine Isothiocyanate with a center wavelength of 542 and FWHM of 20 nm) filters and the Qimaging Retiga 6 camera.

**Flow Cytometry.** Flow cytometry was performed with the BD FACSymphony ™ A1 Cell Analyzer (BD Biosciences, San Jose, CA, USA). For every sample, unstained cultures containing 1% dimethyl sulfoxide (DMSO, Sigma-Aldrich) were used to determine the autofluorescent signal considering size (FSC), complexity (SSC), and fluorescence. The cell viability was determined with Propidium iodide (PI-Sigma-Aldrich, 0.2 µg/mL), while metabolic activity was measured with Thioflavin-T (Thio-T Thermoscientific, Waltham, MA, USA, 30 µg/mL). A total number of 100,000 events were analyzed if not specified otherwise. The data were analyzed using FlowJo (version 10.7.2, FlowJo LLC, Ashland, Jackson County, OR, USA) software.

**RNA extraction and quantitative real-time PCR.** For gene expression, the antibiotic treatment was performed for 15 min in the exponential or stationary phase. The extraction of the total RNA and qRT-PCR was carried out using the method previously outlined [35]. Briefly, *C. difficile* cultures were treated with phenol acid to extract the total RNA from the culture. Subsequently, 1 µg of total RNA was converted into cDNA through reverse transcriptase and random primers (Promega, Madison, WI, USA). These cDNAs, in turn, were employed to carry out the qRT-PCRs reaction in the following manner: a 10 µL reaction containing 25 ng cDNA, 5 µL of Brilliant II SYBR Green QPCR Master Mix (Agilent Technologies, Santa Clara, CA, USA), 0.25 µM of each primer (Appendix A), and water. The PCR conditions consisted of 50 °C for 2 min, 95 °C for 10 min, followed by 95 °C for 10 s, 60 °C for 30 s, and 72 °C for 30 s for a total of 40 cycles. Melting curves were created by increasing the temperature by 1 °C within 60 °C to 95 °C. The data derived from the real-time PCR were analyzed, employing 16s rRNA and *dnaK* levels for normalization, following a previously detailed methodology [35]. The generation of the graphics was accomplished through the utilization of GraphPad Prism 7 software. The experiment was conducted with three biological replicates, each with three technical replicates.

## 3. Statistical Analysis

GraphPad software was employed for graphing and analyzing the data using statistical assays for non-parametric data, such as a one-way and two-way ANOVA, accompanied by Sidak and Bonferroni posthoc tests.

## 4. Results

### 4.1. Enrichment of Spontaneous Persister Cells of C. difficile through Lysis Treatment

Distinguishing between triggered and spontaneous persisters has been considerably challenging, hindering our ability to assess the role of antibiotics, particularly vancomycin, in generating tolerance through the formation of persister cells [21,22,23,24,32]. In this context, we aimed to evaluate a lysis method involving weakening the cell wall of actively dividing cells. Given that persister cells do not divide, we expect that this approach will enable us to reduce the population of non-persister vegetative cells, facilitating subsequent studies with persister cells.

In order to distinguish between triggered and spontaneous persister cells, an enrichment process was conducted utilizing a lysis treatment (LT) during both the exponential and stationary growth phases (Figure 1). A Δ*spo0A* mutant strain was used to confirm if the observed phenomena were solely due to vegetative cells, not spores [32]. In the exponential phase, the plateau of persisters was achieved with just 10 min of treatment for both the WT and Δ*spo0A* strains (Figure 1A), resulting in a noticeable delay in cell death for both strains. Subsequently, the strains collected at the end of the period were revived and subjected to the LT once again, leading to the observation of typical persister behavior replicated in the resurrected bacteria. Similarly, in the stationary phase, a delay in bacterial death was observed after 10 min of treatment. A sustained decrease of 1 to 1.5 orders of magnitude was observed for each strain (Figure 1B). This indicates that the lysis treatment enables culture enrichment, with cells exhibiting a classic biphasic curve during the exponential and stationary phases, which was obtained significantly faster than observed in previous studies [32].

The key characteristics of persister cells include reduced metabolism and the ability to halt or decrease cell division [17]. Previous studies have suggested that ribosome inactivation (resulting in reduced RNA metabolism or the inhibition of protein synthesis) leads to a dormant state (non-dead) [36,37]. To assess if metabolism was diminished, we evaluated the activity of RNA metabolism (activated ribosomes) and determined the viability of these bacteria. Thioflavin-T (Thio-T) has been demonstrated to serve as an RNA-binding probe, enabling the monitoring of RNA metabolism to distinguish between normal and persister cells [33]. To evaluate the metabolic activity of the bacteria isolated using lysis treatment, Thio-T staining was employed to assess RNA metabolism, while Propidium iodide (PI) staining was utilized to determine cell viability. As depicted in Figure 2A, during the exponential phase, the bacteria isolated via lysis treatment were predominantly viable (no PI staining). Still, they exhibited decreased RNA metabolism, strongly indicating their classification as persister bacteria. Interestingly, upon examination of the bacteria isolated using lysis treatment during the stationary phase, it was observed that a significant portion of the culture displayed reduced RNA metabolism even without lysis treatment (Figure 2B), suggesting the presence of spontaneous persister cells. Notably, many of these cells were stained with PI, indicating a mixture of cells with active RNA metabolism, inactive RNA metabolism, and dead cells. However, upon application of the lysis treatment, the culture exhibited inactive RNA metabolism, with no dead cells detected via PI staining, thereby confirming the enrichment of spontaneous persister cells during the stationary phase (Figure 2B). The viability of these cells was confirmed by plating at the end of the experiment (Figure 3B,C). One aspect we did not expect to observe was the decrease in the size of the bacilli observed when enriching the culture with spontaneous persister cells, which was evident in both the exponential and stationary phases (Figure 2). These findings suggest that the lysis treatment effectively enriches the culture with spontaneous persister cells, characterized by reduced RNA metabolism indicative of ribosome inactivation, while maintaining cell viability.

### 4.2. Induction of Triggered Persister Cells of C. difficile Using Vancomycin

Antibiotic therapies play a crucial role in the triggering of CDI, involving the use of cephalosporins, fluoroquinolones, and rifampicin [3]. While these antibiotics are commonly prescribed for other conditions, they inadvertently disrupt the microbial balance in many patients, predisposing them to *C. difficile* infection via spore transmission [4,5,6]. On the other hand, vancomycin, recognized as the first-line therapy for treating *C. difficile*, effectively alleviates symptoms in most patients, although the recurrence of infection may occur in some instances [2,38,39].

Therefore, to assess whether the antibiotics encountered by this pathogen during its infective cycle [3,4,5,6] would induce the formation of persister cells, we selected antibiotics from the same classes to evaluate their ability to induce the formation of triggered persister cells, vancomycin, ceftriaxone, pefloxacin, and rifampicin. The study findings revealed that a one-hour treatment with antibiotics at 10× the minimum inhibitory concentration (MIC) increased the survival rates to lysis treatment by up to 10% (for vancomycin) compared to cases without this antibiotic (Figure 3A). No survival was observed for rifampicin. Considering this, we chose vancomycin for all experiments. Additionally, it was determined that induction with vancomycin followed by enrichment with lysis treatment did not affect the MIC against this antibiotic (10 μg/mL). The viability of these cells was confirmed by plating at the end of the experiment (Figure 3B, exponential and Figure 3C, stationary). This suggests a potential persister behavior.

To evaluate the persister behavior triggered by vancomycin induction, we investigated different enrichment times with lysis treatment in both the exponential and stationary phases (Figure 4). During the exponential phase, we observed a typical persister response, where induction followed by isolation halted cell death after 2 min of treatment (Figure 4A). Although the number of cells collected decreased after the lysis treatment in the procedure with revived strains, we still observed persister behavior and a cessation of death (Figure 4B).

This indicates that the lysis treatment allowed us to enrich the culture with the vancomycin-induced triggered persister phenotype.

To assess whether these triggered persister cells with the persister phenotype exhibited reduced metabolism and were alive [14,17], we examined the RNA metabolism and membrane permeability using the Thio-T and PI stains of collected cells. During the exponential phase with vancomycin, the cells revealed different fluorescent intensities, which persisted, albeit in reduced numbers, after the lysis treatment. This suggests that vancomycin treatment modifies the expression of stress-related genes due to the high concentrations of vancomycin, a phenomenon also observed in the stationary phase (Figure 5). Furthermore, we found that this culture enriched in triggered persister cells exhibited a mixture of sizes (long and short bacilli) in the exponential phase (Figure 5A, LT) and only long bacilli in the stationary phase (Figure 5B, LT). These results suggest that the lysis treatments effectively enrich the culture with vancomycin-induced triggered persister cells, which exhibit different sizes of bacilli (see Appendix A). Additionally, they indicate that within vancomycin-induced triggered persister cells, not all exhibit inactivated ribosomes, as evidenced by the different fluorescent intensities.

### 4.3. Flow Cytometry Analysis of Enriched Cultures Containing Persister Cells in the Exponential Growth Phase

One of the greatest challenges thus far has been conducting measurements in the presence of oxygen with *C. difficile*, given its nature as an obligate anaerobe. We decided to evaluate whether the observations made with LT and vancomycin could be replicated using flow cytometry, employing the same staining techniques. Notably, a differential behavior between the positive control and the treatments was observed during these experiments (Figure 6). Initially, populations were highly defined during the exponential phase, with more than 40% of bacteria showing reduced metabolism, as indicated by Thio-T staining. This can be observed in the lower left quadrant called DN (Figure 6A). This suggests a shift towards the induction of other populations, such as sporulation or persister cells. Remarkably, vancomycin treatment increased this phenomenon to 60%, a percentage that remained consistent when using the LT (Figure 6A). Additionally, it was observed that most of the Thio-T-stained population was stained with PI, strongly suggesting that these cells had damaged walls or were dead, likely due to oxygen and/or the treatments used (Figure 6A). As shown in Figure 6C, this leads us to speculate that the population with the peak on the right of the black line was dead or damaged cells. Finally, when comparing the double-negative population (persister cells), we identified two populations called P1 and P2, as shown in Figure 6B (upper panel). In contrast, the double-positive population (bacteria with active metabolism and damaged cell walls or dead) showed an extended population without clear differences in size and complexity (Figure 6B, lower panel). This observation is consistent with the vancomycin treatment results seen under microscopy, showing both long and short bacilli (Figure 5A). Unfortunately, when replicating this procedure in the stationary growth phase, we could only observe bacteria strongly stained with PI, indicating that 99% of the culture was either dead or had damaged cell walls.

### 4.4. Evaluation of the Expression of Persister-Related Genes

For numerous years, the implication of specific genes in the formation of persister cells has been demonstrated, suggesting mechanisms associated with their functionality [17]. Toxin–antitoxin (TA) systems serve as a prime example of this phenomenon, as it has been proposed that these toxins, when overexpressed, hinder bacterial growth, thus reducing cell division and metabolism in the bacteria under investigation [27,28,29,30,31]. One mechanism for activating these toxins involves the degradation of antitoxins, facilitated by specific proteases belonging to the Lon and Clp families [20,40].

Based on previous data, the study aimed to assess gene expression changes in persister cell formation. Genes encoding MazF, RelE, Cog, and Fic were evaluated [33], along with Lon, ClpP1, ClpP2, ClpC, ClpX, and ClpB. The expression levels of the cog and fic toxins were undetectable in the exponential phase, while relE and mazF were suppressed by the treatments (Figure 7A). Chaperone genes showed increased expression with vancomycin and lysis treatment, indicating involvement in triggered persister cells. ClpX had increased expression with vancomycin, suggesting a relationship with stress, while clpC decreased with lysis treatment. ClpB showed complex regulation patterns (Figure 7B).

Peptidase *clpP1* had increased expression with vancomycin and decreased with lysis treatment. As for *clpP2*, an increase in expression was noted with vancomycin treatment and vancomycin-induced persister cells, along with a decrease only with the lysis treatment, suggesting a role in the formation of persister cells induced by vancomycin. In the case of *lon*, a decrease in expression was observed with the vancomycin treatment and vancomycin-induced persister cells, further suggesting non-involvement in this phenomenon (Figure 7C).

TA system toxin genes behaved differently in the stationary phase, with mazF increasing during the vancomycin and lysis treatment (Figure 7D). For *relE*, an increase was only observed with the vancomycin treatment and vancomycin-induced persister cells (Figure 7D). This suggests that *relE* is involved in the response to vancomycin-induced stress and in the induction of triggered persister cells but not in spontaneous persister cells. Significantly, the *fic* and *cog* expression levels were elevated in the cells induced to persist by the vancomycin treatment and in the cells treated exclusively with vancomycin in the stationary phase (Figure 7D). Regarding the expression of chaperone genes *clpC*, *clpX*, and *clpB*, no significant differences were observed in the expression of *clpB*. However, *clpX* showed increased expression in the cells treated with vancomycin and vancomycin-induced persister cells. Furthermore, *clpC* demonstrated a substantial increase in expression in vancomycin-induced persister cells and bacteria treated solely with vancomycin (Figure 7E). For *lon*, there was consistently maintained reduced expression in all three treatments, and no significant differences were observed in *clpP1*. *clpP2* displayed an increase in expression in vancomycin-induced persister cells and a lower increase with vancomycin (Figure 7F). These results suggest that the toxins MazF and RelE do not directly participate in the induction of spontaneous persister enrichment, and all three chaperones, ClpC, ClpX, and ClpB with ClpP2 induce the formation of the triggered persister phenotype in the exponential phase. Moreover, the four toxins of TA systems, MazF, RelE, Fic, Cog, and chaperones ClpC and ClpX with ClpP2 participate in the induction of the vancomycin-induced triggered persister phenotype, and only MazF, Fic, and ClpC induce the formation of spontaneous persisters cells in *C. difficile.*

## 5. Discussion

Persister cells typically accumulate during the exponential growth phase, reaching their peak expression in the stationary phase, which aligns with the characterization of spontaneous persister cells [17]. This observation was supported by research from Cañas-Duarte in 2014, indicating that such cells arise stochastically during bacterial growth [21].

This study demonstrated the presence of both triggered persister cells induced by vancomycin and spontaneous persisters in *C. difficile*. It is worth noting that the presence of taurocholate, acting as a spore germinant, initiated the germination process in the spores within the culture, leading to a decrease in the total number of spores in the medium, consistent with findings from previous research by other investigators [41]. Additionally, rapidly metabolized sugars, such as glucose and fructose, have been found to regulate approximately 18% of the *C. difficile* transcriptome, with half of these genes regulated by CcpA. This protein forms part of a regulon responsible for controlling sugar uptake and gene regulation [42]. Earlier studies have demonstrated that CcpA suppresses the expression of Spo0A and SigF, which are early regulators of sporulation [42]. Consequently, the presence of these sugars inhibits the expression of these genes, thus preventing spore formation during cultivation, which corroborates the absence of spores in all the cultures analyzed. The lysis treatment unveiled a plateau of persister cells, indicating the presence of cells with reduced metabolism and cell division [21]. A comparison with a previous study by Cañas-Duarte in 2014 revealed consistency in identifying the plateau within a similar time interval despite the lack of detail regarding the lysis treatment in that study. Unlike Cañas-Duarte, our approach involved generating a curve with revived cells, thereby resolving potential confusion between persister and resistant cells.

Thioflavin-T and Propidium Iodide stains helped us observe *C. difficile* cells with inactive RNA metabolism and living cells after lysis treatment. This approach provided an alternative to the cell sorting methods used for aerobic bacteria and revealed unique shapes in spontaneous persister cells [21], which appeared as short bacilli. Additionally, when examining the complexity of the cells using flow cytometry, we found that most cells treated with the LT had fewer complex bacilli. However, with vancomycin and vancomycin combined with the LT, a new population of more complex cells emerged (Figure 6B). We believe these more complex cells are the long and short bacilli seen under the microscope. This finding offers new insights into *C. difficile* and adds to the growing understanding of the different shapes and electron densities of persister cells [37].

The antibiotic evaluation showed that vancomycin and pefloxacin increased the number of persister cells, with the former selected for further investigation due to its use in treating *C. difficile*-associated diseases. This finding supports previous studies suggesting the involvement of vancomycin in the persister cell formation [32].

In order to demonstrate the persister phenotype, bacteria were revived, and the protocol was replicated. This phenomenon mirrors observations conducted by other researchers using *E. coli* as a model, where persister cells were isolated with antibiotics (without lysis treatment) [17], and upon bacterial revival, the phenomenon recurred. Importantly, this consistency in occurrence appears independent of the antibiotic utilized, indicating that methods for isolating persister cells are generally unaffected by cells acquiring antibiotic resistance. Notably, these observations were made during the exponential phase, indicating the predominance of triggered persister cells.

Upon the microscopic examination of these cultures, various bacterial morphologies were discernible. In the phase contrast, the elongated bacillus morphology predominated, alongside a significant presence of bacteria with the short bacillus morphology.

Thioflavin-T and Propidium Iodide stains helped us observe *C. difficile* cells with inactive RNA metabolism and living cells after lysis treatment. This approach provided an alternative to the cell sorting methods used for aerobic bacteria and revealed unique shapes in spontaneous persister cells, which appeared as short bacilli.

The examination of the bacteria treated solely with vancomycin (VAN) revealed that many exhibited decreased RNA metabolisms but remained predominantly viable, with no apparent cell wall damage (Figure 6A). However, the observation of the vancomycin treatment followed by the lysis treatment revealed a mixture of morphologies, with numerous cells displaying activated RNA metabolism, suggesting a diverse bacterial population in the isolated subculture. These populations were observed through microscopy and flow cytometry, where differences in bacillus length, RNA metabolism, and bacterial complexity were noted (Figure 5). Furthermore, when examining the complexity of the cells using flow cytometry, we found that most cells treated with the LT had fewer complex bacilli. However, with vancomycin and vancomycin combined with the LT, a new population of more complex cells emerged. We believe these more complex cells are the long and short bacilli seen under the microscope (Figure 5A, LT with zoom). This finding offers new insights into *C. difficile* and adds to the growing understanding of the different shapes and electron densities of persister cells. Studies conducted with other models, such as *Salmonella enterica*, have demonstrated that the persister subculture encompasses bacteria with arrested cell division but activated metabolism, termed viable but non-culturable (VBNC) [19]. While these cells typically do not contribute to culture resurrection, they play roles in responses such as biofilm formation [43]. In the context of *C. difficile*, research suggests that vancomycin treatments induce biofilm formation in the stationary phase [44], which could elucidate the observed results, as biofilm formation involves alterations in gene expression. Within these cellular subcultures, division ceases, but metabolism persists to facilitate the formation of these structures [43].

In the assessment of gene expression, an unexpected finding emerged: during the exponential phase, the TA system toxin genes *relE* and *mazF* were repressed under the applied treatments, while *fic* and *cog* were undetectable. Conversely, in the stationary phase, all four toxins were overexpressed. This suggests that the behavior of these TA systems mirrors that observed in other bacteria, where they are activated or overexpressed during stress-induced persister cell formation [27,29,31,45].

In the case of chaperones, during the exponential phase, *clpC* and *clpB* were implicated in forming spontaneous persister cells, while all three chaperones played a role in developing triggered persister cells. In the stationary phase, only *clpC* and *clpX* were notable for their involvement in forming triggered persisters, with *clpC* implicated in both types. Regarding peptidases and proteases, *clpP2* emerged as the most intriguing candidate, exhibiting altered expression under treatments in both growth phases. This finding is compelling because Clp family chaperones and peptidases are closely linked to the activation of type II TA systems [46,47,48], suggesting their joint participation, which warrants further investigation in subsequent studies. Additionally, it was demonstrated that changes in chaperone expression influence the targets of Clp protease complexes, regulating, for example, the abundance of misfolded proteins (or total proteins) in the cell, a factor associated with persister cells [15,18].

In summary, this study not only validates the existence of persister cells in *C. difficile* induced by vancomycin but also sheds light on new perspectives regarding the morphology of these cells, their response to treatments, and their capacity to induce the formation of triggered persister cells. These findings enhance our comprehension of persistence of *C. difficile* and offer valuable insights for future research and therapeutic approaches.

## Figures and Tables

**Figure 1 microorganisms-12-01394-f001:**
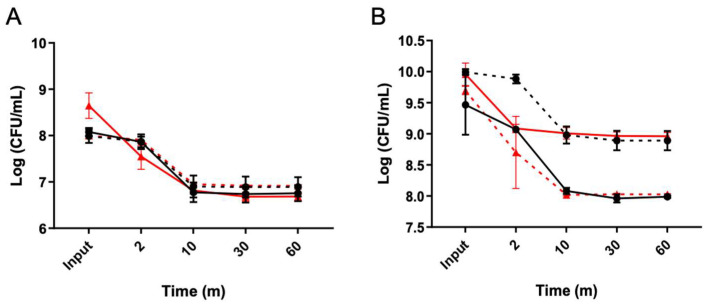
*C. difficile* persisters cells enriched with a lysis treatment. The lysis treatment was conducted after the culture reached OD 0.5 ((A), exponential) with aliquots collected at 2, 10, 30, and 60-minute intervals. Colonies emerging from the 60 min seeding were further cultivated for 16 h, and the experiments were repeated, as depicted in (**A**). Another aliquot was taken from the overnight culture to grow to OD 0.9 to seed the stationary phase (**B**), and the lysis treatment was performed, with the aliquots sampled at the same time intervals for each exponential assay. The colonies obtained from the 60 min seeding were also regrown for 16 h, and the experiments were repeated. The presence of persister cells reached a plateau starting from the 10 min point onwards in both cases (**A**,**B**). Wild type (black circles), resurrected wild type (black squares, dashed line), Δ*spo0A* (red triangles), resurrected Δ*spo0A* (red triangles, dashed line). Statistics: a one-way ANOVA with the Bonferroni posthoc test; no significant differences were found when adjusting the survival values with each input (*n* = 3).

**Figure 2 microorganisms-12-01394-f002:**
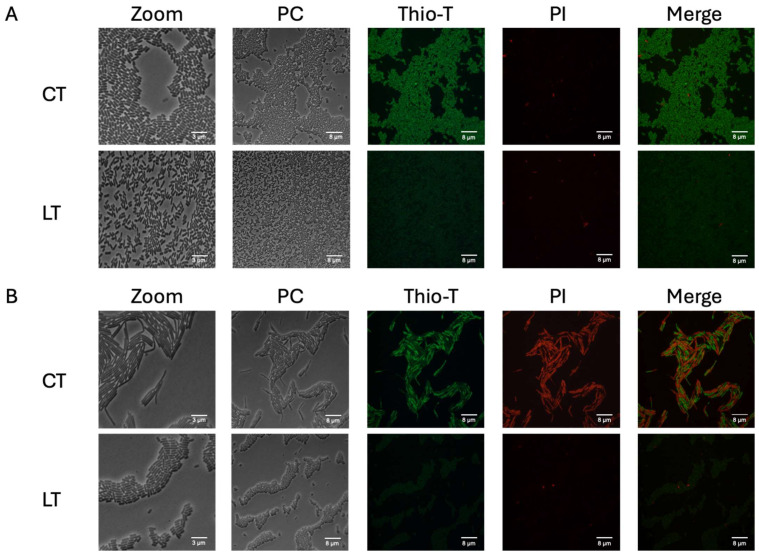
RNA metabolism of enriched spontaneous persister cells. An aliquot extracted from the culture during the exponential phase (**A**) was used for the experiment, both untreated (CT, control treatment) and following a 15 min lysis treatment (LT, lysis treatment). The presence of green fluorescence indicated active RNA metabolism, while the absence of color indicated inactive RNA metabolism. Cell wall damage and cell death were evaluated using propidium iodide (PI) staining, shown in red. The same methodology was applied to a culture in the stationary phase (**B**) with both CT and LT treatments. PC: phase contrast; Thio-T: Thioflavin-T stain; PI: Propidium iodide stain. Each experimental condition was replicated three times (*n* = 3).

**Figure 3 microorganisms-12-01394-f003:**
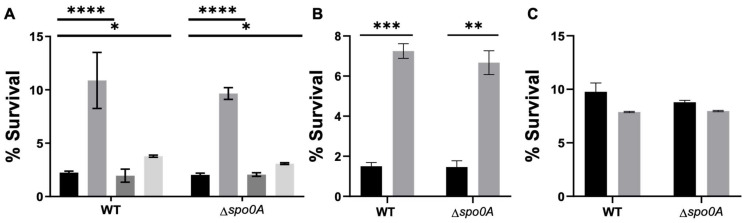
Antibiotic treatment and persister cell survival enriched with an LT. (**A**) A survival assay was conducted during the exponential phase, involving a 1 h antibiotic treatment with vancomycin (VAN, grey), ceftriaxone (CFX, dark grey), or pefloxacin (PEF, light grey). Each antibiotic was used separately, and after the treatment, an aliquot was plated to mark the input of the experiment. Subsequently, a 15 min lysis treatment was applied. (**B**,**C**) Percentage of survival of *C. difficile* WT or Δ*spo0A* after 10× VAN treatment. % of survival of *C. difficile* strains after no treatment (CT, black) or vancomycin (10× VAN, grey) in the exponential (**B**) or stationary phase (**C**). The experiment was replicated three times (*n* = 3), and the statistical analysis was performed using a one-way ANOVA with posthoc Sidak’s test (* *p* < 0.05; ** *p* < 0.005; *** *p* < 0.0005; **** *p* < 0.0001).

**Figure 4 microorganisms-12-01394-f004:**
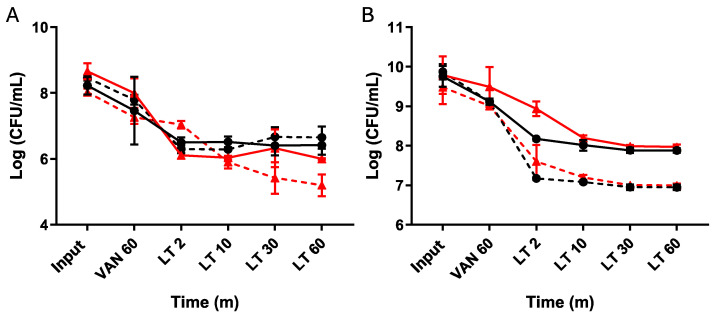
Vancomycin-induced persister cells enriched with an LT. Cultures of *C. difficile* were grown overnight (O/N), and an aliquot was transferred to reach the exponential phase (**A**) following the established procedure. At the experiment’s outset, an aliquot was seeded (input), followed by a 1 h treatment with vancomycin before seeding again. Subsequently, a 60 min lysis treatment was administered, with samples collected at 2, 10, 30, and 60 min. The same experimental setup was repeated, starting from the overnight culture (**B**) with aliquots seeded for the input, followed by one hour of vancomycin treatment, then subjected to 2, 10, 30, and 60 min of lysis treatment. The plateau of persister cells was reached after 10 min of treatment. Resurrected bacteria from the 60 min lysis treatment aliquot were retrieved and used to repeat the experiment. Wild type (black circles), resurrected wild type (black squares, dashed line), Δ*spo0A* (red triangles), resurrected Δ*spo0A* (red triangles, dashed line). Statistics: a one-way ANOVA with the Bonferroni posthoc test; no significant differences were found when adjusting the survival values with each input (*n* = 3).

**Figure 5 microorganisms-12-01394-f005:**
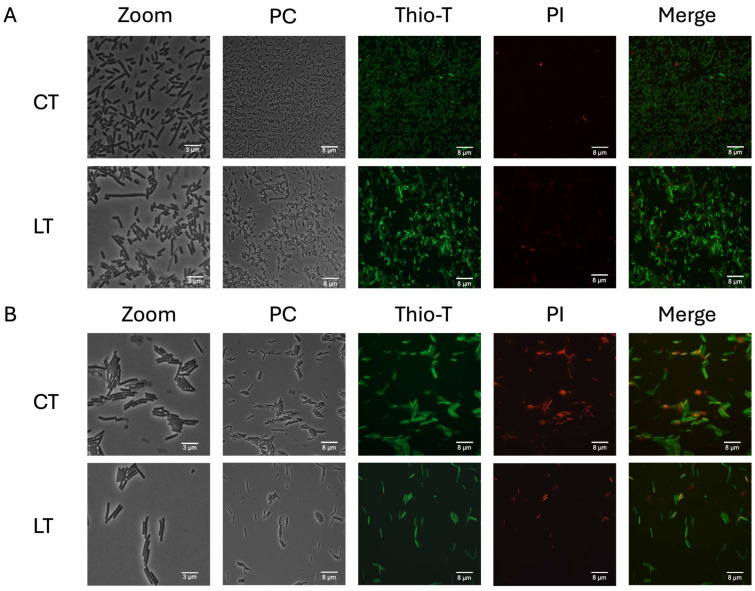
RNA metabolism of enriched triggered persister cells under vancomycin treatment. Panel (**A**) illustrates the exponential phase growth culture treated for 1 h with vancomycin with and without lysis treatment (LT) or the control treatment (CT). A similar experiment is depicted in the stationary phase (panel (**B**)). Thioflavin-T (Thio-T) staining was utilized to visualize active RNA metabolism (green color) or inactive metabolism (pale green staining), while propidium iodide (PI) staining was employed to observe cell wall damage or cell death, represented in red. In both experiments, a mixture of bacteria with active and inactive metabolism is observed, with the exponential phase exhibiting more cells with active RNA metabolism. Conversely, in the stationary phase, the CT condition reveals more cells with damaged cell walls or dead cells. PC: phase contrast; Thio-T: Thioflavin-T stain; PI: Propidium iodide stain. Each experimental condition was replicated three times (*n* = 3).

**Figure 6 microorganisms-12-01394-f006:**
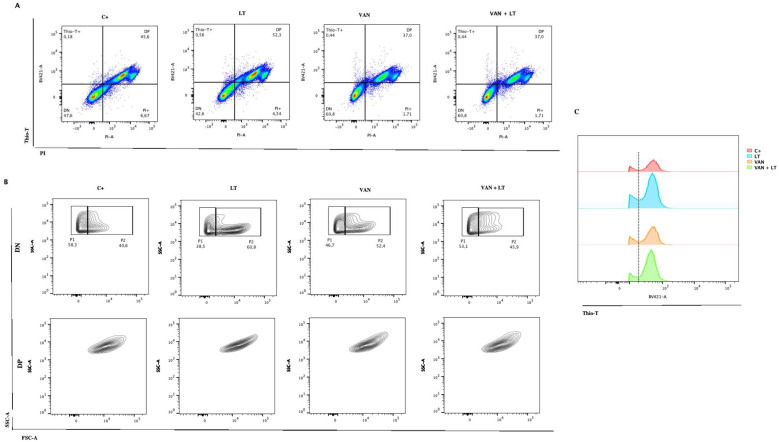
Characterization of persister *C. difficile* cells with flow cytometry. (**A**) Identification of *C. difficile* stained with Thio-T and PI after application of the different treatments: lysis treatment (LT), vancomycin (VAN), vancomycin + lysis treatment (VAN + LT), (C+) positive control of growth. (**B**) Identification of the size and complexity of double-negative (DN) (upper) and double-positive (DP) (down) populations of *C. difficile* stained with Thio-T and PI. (**C**) A representative histogram of total Thio-T expression in *C. difficile* was previously treated with C+, LT, VAN, and VAN + LT. The black dotted line shows the negative signal for Thio-T to the left and the positive signal for Thio-T to the right of the line.

**Figure 7 microorganisms-12-01394-f007:**
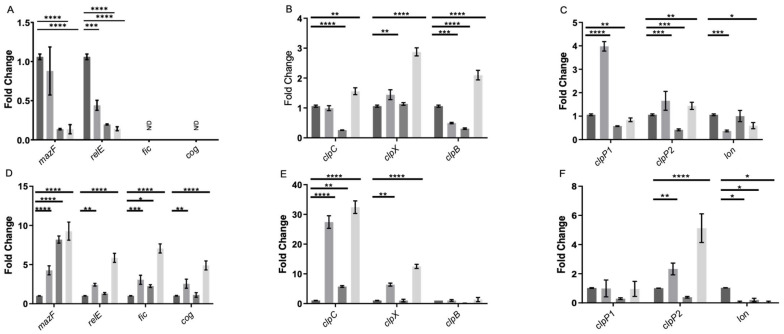
Gene expression of enriched persister cells under vancomycin treatment. RNA extraction was performed during the exponential growth phase (**A**–**C**) and the stationary phase (**D**–**F**). In each growth phase, the gene expression was evaluated under four conditions: control (dark grey), a 15 min treatment with vancomycin (VAN 15, grey), a 15 min lysis treatment (LT 15, carbon grey), and a sequential treatment involving vancomycin followed by a 15 min lysis treatment (VAN 15 LT, light grey). The expression levels of toxin genes from TA systems, namely *relE*, *mazF*, *fic*, and *cog* (**A**,**D**), chaperone genes from the *clp* family, including *clpC*, *clpX*, and *clpB* (**B**,**E**), as well as peptidase and protease genes, specifically *clpP1*, *clpP2*, and *lon* (**C**,**F**) were measured. The experiment was conducted with *n* = 3 replicates, and the statistical analysis was performed using a one-way ANOVA with posthoc Bonferroni’s test (* *p* < 0.05; ** *p* < 0.005; *** *p* < 0.0005; **** *p* < 0.0001).

## Data Availability

The original contributions presented in the study are included in the article/Appendix A, further inquiries can be directed to the corresponding authors.

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
