# Peer review of "Characterization of *Clostridioides difficile* Persister Cells and Their Role in Antibiotic Tolerance"

_microorganisms, 2024, doi:10.3390/microorganisms12071394_

Round 1
Reviewer 1 Report
Comments and Suggestions for Authors
Authors differentiated between two distinct types of persister cells, namely triggered persister cells induced by vancomycin during both the exponential and stationary phases and spontaneous persister cells induced by stochastic processes. Additionally, they identified RNA metabolic-decreased persister cells in C. difficile. In addition to these findings, they have also observed overexpression of toxin-antitoxin systems and Clp protease genes, which are known to play pivotal roles in forming persister cells in several bacteria. The experiments are well designed to identify persister cells.
Specific Comments:
1) Lines in Fig 1 is hard to distinguish, need improvement. The figure legend description is not clear to follow and understand.
DO?
2) Fig 3, RIF is not in figure.
3) Fig 6B upper panel: DP /DN?? population? confusing
Comments on the Quality of English Language
The 4.1 section, second paragraph content including Fig 1 (In order to....) is hard to follow
Author Response
Dear Reviewer 1:
Attached you will find answers to your observations
Comments and Suggestions for Authors
Authors differentiated between two distinct types of persister cells, namely triggered persister cells induced by vancomycin during both the exponential and stationary phases and spontaneous persister cells induced by stochastic processes. Additionally, they identified RNA metabolic-decreased persister cells in C. difficile. In addition to these findings, they have also observed overexpression of toxin-antitoxin systems and Clp protease genes, which are known to play pivotal roles in forming persister cells in several bacteria. The experiments are well designed to identify persister cells.
Response: Thank you very much for these comments. We are very satisfied with our results, considering that this is a first characterization of spontaneous and triggered persister cells in C. difficile.
Specific Comments:
1) Lines in Fig 1 is hard to distinguish, need improvement. The figure legend description is not clear to follow and understand.
Response: Thanks for this observation. Fig 1 was improved
2) Fig 3, RIF is not in figure.
Response: Thanks for this observation. RIF result was added in text
3) Fig 6B upper panel: DP /DN?? population? confusing
Response: Thanks for this comment. This was our mistake. New Figure 6B was added
Comments on the Quality of English Language
The 4.1 section, second paragraph content including Fig 1 (In order to....) is hard to follow
Response: Done, this was improved

Reviewer 2 Report
Comments and Suggestions for Authors
Figures and tables are poorly labeled and inadequately referenced, impeding comprehension. The statistical analysis is superficial, with no clear explanation of the chosen tests.
The results are weak and do not convincingly support the claims made. Interpretations are speculative and lack robust experimental evidence.
The manuscript requires substantial revisions. Improved experimental design, rigorous data analysis, and a clearer articulation of novel contributions are essential. Currently, the study does not meet the standards for publication in a scientific journal.
Author Response
Dear Reviewer 2:
Attached you will find answers to your observations
Comments and Suggestions for Authors
Figures and tables are poorly labeled and inadequately referenced, impeding comprehension. The statistical analysis is superficial, with no clear explanation of the chosen tests.
Response: Thanks for this comment. All figures were improved and well referenced. Statistical test used were described in methods section and in legends of figures
The results are weak and do not convincingly support the claims made. Interpretations are speculative and lack robust experimental evidence.
Response: Thanks for this comment. Respectfully we are in disagree with the reviewer. All experiments were performed according to literature evidence with improvements according to our study model. Below we show some related articles that support our experimental design and results: doi:10.1016/j.jgar.2023.05.004; doi:10.1007/s10541-005-0111-6; doi:10.1371/journal.pone.0088660; doi:10.1534/genetics.104.035352; doi:10.1186/1471-2180-14-120; doi:10.1007/978-1-4939-2854-5_7
Next we show a brief explanation of every figure:
- Figure 1 and 4 show clearly biphasic curves characteristics of persister behavior and demonstrated that persisters are not spores.
- Figures 2 and 5 show fluorescence microscopy in which RNA metabolism are decreased in persister cells in both exponential and stationary growth phases are clearly observed.
- Figure 3 shows % of survival against different antibiotics. Considering this result we choose vancomycin from other experiments
- Figure 6 is a flow cytometry that shows the behavior of persister cells and how populations change in percentage after treatments
- Figure 7 shows expression of several genes described in literature which are known to play pivotal roles in forming persister cells in several bacteria
- CT: Control without treatment, in which RNA extraction was performed from all bacteria
- LT: Lysis treatment, in which RNA extraction was performed only from survivor bacteria after lysis (spontaneous persisters)
- VAN: Vancomycin treatment, in which RNA extraction was performed from all bacteria
- VAN + LT: Vancomycin induction and lysis treatment, in which RNA extraction was performed only from survival bacteria after induction and lysis (triggered persisters)
With these results we strongly suggest differential expression of persister genes
The manuscript requires substantial revisions. Improved experimental design, rigorous data analysis, and a clearer articulation of novel contributions are essential. Currently, the study does not meet the standards for publication in a scientific journal.
Response: Thanks: all manuscript was improved according to the observations of all reviewers

Reviewer 3 Report
Comments and Suggestions for Authors
This manuscript by Inostroza et al discusses the use of an assay that selectively kills off/lyses dividing cells in culture to define a sub-population of C. difficile organisms called persisters. These cells appear to be metabolically inactive but not dead based on staining characteristics. Interestingly , these are not spores, as they develop comparably in cultures of non-sporulating (SpoOA negative) C.difficile at similar rates. They occur in both log and stationary phases of growth -with increased prevalence among the latter.
They are also induced above baseline rates by exposure to vancomycin, and are functionally resistant to killing by vancomycin -thus “persisting” in culture. When induced to grow in fresh media , they become mostly susceptible to vancomycin but a fraction again re-enters the “persister state”.
The clinical relevance of these persisters is not explored, though its mentioned that they may contribute to relapse of disease and possibly emergent drug resistance.
The work is well-written and well-presented. I have minor points I think the authrs should correct/address:
Introduction
1st paragraph-define FMT;
-in last sentence, “this antibiotic” is ambiguous -change to “vancomycin….”
3rd paragraph-Remove “On the other hand”
Figure 1- clarify triangles in Figure B
Figure 1 legend- “culture wer reached DO) -should be culture reached OD”
-“to growth until reached DO” should be “grown to OD”
-“plateau starting at 15 minutes” should be “….at 10 minutes”
Results page 5 – “Principio del formulario” is misplaced-rmove this
Figure 2B-in the photo, individual bacterial cells are not seen-just a large clump of biomaterial. Is this representative of the entire sample?
Figure 3A -can be eliminated -state this result in the text
Figure 3B -is there data for RIF or is it just missing?
Page 6 bottom –“halted cell death after 10 min” should be “..after 2 minutes”
Page 7 2nd paragraph –“mixed intensity” is not clear- did you mean dual staining.
Section 4.4-is too long-winded. The text should not just re-iterate the findings in the figure but try to synthesize/summarize the findings. Paragraph 3 can be split into 3 paragraphs – dealing with peptidases, toxin genes and chaperones.
Figure 7 -point out tha the axes are different for each componenet mini-graph-can point this out in the text.
Discussion 2nd paragraph -after “spontaneous” ad “Persisters”
Fig S2 , S3 and S4 are compelling and should be in the main text, not supplements. Of course, some more comment related to them would have to be introduced in methods, results and discussion.
Comments on the Quality of English Languagewell-written
Author Response
Dear Reviewer 3:
Attached you will find answers to your observations
Comments and Suggestions for Authors
This manuscript by Inostroza et al discusses the use of an assay that selectively kills off/lyses dividing cells in culture to define a sub-population of C. difficile organisms called persisters. These cells appear to be metabolically inactive but not dead based on staining characteristics. Interestingly , these are not spores, as they develop comparably in cultures of non-sporulating (SpoOA negative) C.difficile at similar rates. They occur in both log and stationary phases of growth -with increased prevalence among the latter.
They are also induced above baseline rates by exposure to vancomycin, and are functionally resistant to killing by vancomycin -thus “persisting” in culture. When induced to grow in fresh media , they become mostly susceptible to vancomycin but a fraction again re-enters the “persister state”.
The clinical relevance of these persisters is not explored, though its mentioned that they may contribute to relapse of disease and possibly emergent drug resistance.
Response: Thank you very much for these comments. We are very satisfied with our results, considering that this is a first characterization of persister cells in C. difficile.
The work is well-written and well-presented. I have minor points I think the authrs should correct/address:
Introduction
1st paragraph-define FMT;
Response: Done
-in last sentence, “this antibiotic” is ambiguous -change to “vancomycin….”
Response: Done
3rd paragraph-Remove “On the other hand”
Response: Done, removed
Figure 1- clarify triangles in Figure B
Response: Thanks for this comment. Fig 1 was improved
Figure 1 legend- “culture wer reached DO) -should be culture reached OD”
-“to growth until reached DO” should be “grown to OD”
-“plateau starting at 15 minutes” should be “….at 10 minutes”
Response: Done
Results page 5 – “Principio del formulario” is misplaced-rmove this
Response: Done, we don’t know what happened here
Figure 2B-in the photo, individual bacterial cells are not seen-just a large clump of biomaterial. Is this representative of the entire sample?
Response: Thanks for this comment. Effectively this is a representative figure of all experiment, we speculate that this could be due to biofilm formation in stationary phase as we show in our previous article: doi:10.1016/j.jgar.2023.05.004
Figure 3A -can be eliminated -state this result in the text
Response: Thanks for this observation, we have eliminated Figure 3A and added the result in text
Figure 3B -is there data for RIF or is it just missing?
Response: Thanks for this observation, figure was improved
Page 6 bottom –“halted cell death after 10 min” should be “..after 2 minutes”
Response: Done
Page 7 2nd paragraph –“mixed intensity” is not clear- did you mean dual staining.
Response: Thanks for this comment. Mixed intensity was change for different fluorescence intensities
Section 4.4-is too long-winded. The text should not just re-iterate the findings in the figure but try to synthesize/summarize the findings. Paragraph 3 can be split into 3 paragraphs – dealing with peptidases, toxin genes and chaperones.
Response: Thanks for this comment. This section was resumed and improved
Figure 7 -point out tha the axes are different for each component mini-graph-can point this out in the text.
Response: Done, figure 7 was improved
Discussion 2nd paragraph -after “spontaneous” ad “Persisters”
Response: Done
Fig S2 , S3 and S4 are compelling and should be in the main text, not supplements. Of course, some more comment related to them would have to be introduced in methods, results and discussion.
Response: Thanks for this comment. We respect the opinion of the reviewer. In this sense, figure S3 was merged with figure 3, but we believe that other supplementary figures are methods or controls of our results, so we respectfully consider that they should be in the supplementary section

Round 2
Reviewer 2 Report
Comments and Suggestions for Authors
Good paper